# Extraction and high-throughput sequencing of oak heartwood DNA: Assessing the feasibility of genome-wide DNA methylation profiling

Federico Rossi[1,13], Alessandro Crnjar[2], Federico Comitani[3,4], Rodrigo Feliciano[5,6], Leonie Jahn[1,7], George Malim[1], Laura Southgate[1], Emily Kay[1,8], Rebecca Oakey[1], Richard Buggs[9,10], Andy Moir[11], Logan Kistler[12], Ana Rodriguez Mateos[5], Carla Molteni[2], Reiner Schulz[1] *

1 Department of Medical and Molecular Genetics, King's College London, London, United Kingdom, 2 Department of Physics, King's College London, London, United Kingdom, 3 Department of Chemistry, University College London, London, United Kingdom, 4 The Hospital for Sick Children, Toronto, Ontario, Canada, 5 Department of Nutrition, King's College London, London, United Kingdom, 6 Division of Cardiology, Pulmonology and Vascular Medicine, University of Dusseldorf, Dusseldorf, Germany, 7 Novo Nordisk Foundation Center for Biosustainability, Technical University of Denmark, Kongens, Lyngby, Denmark, 8 CRUK Beatson Institute, Glasgow, United Kingdom, 9 Department of Natural Capital and Plant Health, Royal Botanical Gardens, Richmond, United Kingdom, 10 School of Biological and Chemical Sciences, Queen Mary University of London, London, United Kingdom, 11 Tree-Ring Services Limited, Mitcheldean, United Kingdom, 12 Department of Anthropology, National Museum Of Natural History, Smithsonian Institution, Washington, DC, United States of America, 13 Department of Experimental Oncology, IEO European Institute of Oncology IRCCS, Milan, Italy

* reiner.schulz@kcl.ac.uk

**Data Availability Statement:** The WGS and WGBS data generated and analysed during the current study were deposited in the NCBI Gene Expression

## Abstract

Tree ring features are affected by environmental factors and therefore are the basis for dendrochronological studies to reconstruct past environmental conditions. Oak wood often provides the data for these studies because of the durability of oak heartwood and hence the availability of samples spanning long time periods of the distant past. Wood formation is regulated in part by epigenetic mechanisms such as DNA methylation. Studies of the methylation state of DNA preserved in oak heartwood thus could identify epigenetic tree ring features informing on past environmental conditions. In this study, we aimed to establish protocols for the extraction of DNA, the high-throughput sequencing of whole-genome DNA libraries (WGS) and the profiling of DNA methylation by whole-genome bisulfite sequencing (WGBS) for oak (*Quercus robur*) heartwood drill cores taken from the trunks of living standing trees spanning the AD 1776-2014 time period. Heartwood contains little DNA, and large amounts of phenolic compounds known to hinder the preparation of high-throughput sequencing libraries. Whole-genome and DNA methylome library preparation and sequencing consistently failed for oak heartwood samples more than 100 and 50 years of age, respectively. DNA fragmentation increased with sample age and was exacerbated by the additional bisulfite treatment step during methylome library preparation. Relative coverage of the non-repetitive portion of the oak genome was sparse. These results suggest that quantitative methylome studies of oak hardwood will likely be limited

Omnibus (GEO) with the accession number GEO: GSE143201. The molecular dynamics simulation data are available at the following link: http://doi.org/doi:10.18742/RDM01-502.

**Funding:** This research was funded by the King's College London Department of Medical and Molecular Genetics, and the UK Genetics Society. This work was also supported by the King's Together Multi and Interdisciplinary Research Scheme: Wellcome Trust Institutional Strategic Support Fund (grant reference: 204823/Z/16/Z). Sequencing was performed at the NIHR Biomedical Research Centres, London, UK. AC and CM acknowledge the UK high performance computing service ARCHER, for which access was obtained via the UKCP consortium and funded by EPSRC grant EP/P022472/1, and the UK Materials and Molecular Modelling Hub, which is partially funded by EPSRC grant EP/P020194/1 for computational resources. The funders had no role in study design, data collection and analysis, decision to publish, or preparation of the manuscript.

**Competing interests:** The authors have declared that no competing interests exist.

to relatively recent samples and will require a high sequencing depth to achieve sufficient genome coverage.

## Introduction

Heartwood corresponds to the inner layers of wood that do not contain living cells [1]. Anatomical and physiological changes in sapwood, the external layers of wood, lead to the formation of heartwood. Carbon dioxide enrichment, formation of tylosis, accumulation of gums and phenolic compounds have been proposed as factors determining the death of sapwood residual living cells and the formation of heartwood [2]. Oak heartwood in particular has been reported as a viable source of archival DNA [3]. Such DNA would provide an opportunity to expand dendroclimatology inferences on tree response to past environmental conditions by including epigenomic features that may convey additional and/or more specific information compared to the macroscopic features used so far [4]. Specifically, genome-wide DNA methylation profiling followed by epigenomic association studies could identify genomic regions (including genes) where DNA methylation changes systematically relate to variations of a particular environmental factor [5]. Moreover, the characterisation of oak heartwood DNA could be applied into processes that require the identification of timber such as the assessment of the provenance of the wood during the trading process, or the identification of timber used in buildings [6, 7]. However, to what extent DNA methylation profiling by sequencing of oak heartwood DNA is feasible, given the amounts and quality of residual DNA and the presence of enzyme-inhibiting compounds, is unknown.

Previous attempts to explain failures of PCR when applied to the amplification of DNA extracted from wood typically have focused on phenolic compounds, which are abundant in heartwood [3, 8, 9]. Phenolic compounds are supposed to irreversibly bind to DNA thus causing amplification impairment through three distinct non-covalent modes: electrostatic interactions, intercalative binding and groove binding [10–13]. DNA polymerase inhibition can be prevented by pre- or post-extraction removal of inhibitors, inactivation of inhibitors and the addition of enzyme activity facilitators [14–19]. Heartwood PCR success is also affected by template length, suggesting DNA fragmentation can impede DNA amplification [3, 20, 21]. Moreover, wood storage conditions and duration can impact DNA amplification by reducing the sample exposure to aerobic conditions and limiting DNA degradation [3, 20, 22]. Despite being challenging, others have reported the successful PCR-amplification of DNA extracted from recent as well as ancient oak heartwood, though only chloroplast DNA could be amplified from ancient samples [3, 23].

In addition to PCR, there have also been reports on the high-throughput sequencing of DNA extracted from ancient wood, specifically waterlogged oak wood (sapwood and heartwood) and pine sapwood [24, 25]. The analysis of the wood DNA reads showed features typical of ancient samples such as deamination at overhanging single strands and depurination-driven fragmentation [24, 25]. Despite being authenticated by the presence of features specific to ancient DNA, the low amount of endogenous DNA reads retrieved from oak heartwood renders questionable whether this tissue can be used for genome-wide genetic and epigenetic studies [24].

In this study, we aimed to establish protocols for the extraction of DNA, the high-throughput sequencing of whole-genome DNA libraries (WGS) and the profiling of DNA methylation by whole-genome bisulfite sequencing (WGBS) for oak (*Quercus robur*) heartwood drill cores

taken from the trunks of living standing trees spanning the AD 1776–2014 time period. We optimised the DNA extraction protocol to limit PCR inhibition, maximise DNA yield and extend the range of heartwood sample age from which a DNA library can be obtained. We also studied the interaction of DNA with a potential PCR inhibitor by using molecular dynamics (MD) simulations to elucidate DNA amplification impairment and rationalise how it could be prevented.

## Materials and methods

### Wood core sampling and preparation

The wood (sapwood plus heartwood) samples analysed in the present study were collected from four oak (*Quercus robur*) trees located at the Horsepool Bottom Nature Reserve, National Grid Reference (NGR) SO 668 165 (Jubilee Road, Mitcheldean, Gloucestershire, GL17 OEE, United Kingdom), using a 2-thread increment borer and a core cross-section of an oak (*Quercus robur* or *Quercus petraea*) tree trunk collected in the Dulwich Woods, NGR TQ 34340 72608 (5 Peckarmans Wood, London SE26 6SB, United Kingdom) (Table 1). No sampling permits were required. The Horsepool Bottom Nature Reserve is privately owned and operated by co-author Dr Andy Moir, who also performed the core sampling. The Dulwich Woods sample was a discarded trunk cross-section, left on the ground after public footpath maintenance. Samples were left to dry for subsequent dendrochronological analysis or stored at -20˚C without being dated (Table 2).

All of the samples were used for the test of DNA extraction. The GLOR drill cores were divided into 5–15 years-long segments using ethanol-wiped scissors. For the cores that were not dated, the division into 5–15 years segments was accomplished using as reference the plot

**Table 1. Girth, National Grid Reference and species of the oak samples.**

| Tree code | Girth (m) | National Grid Reference | Species |
|---|---|---|---|
| GLOR01 | 4.25 | SO 6679 1658 | *Quercus robur* |
| GLOR02 | 4.70 | SO 6675 1653 | *Quercus robur* |
| GLOR03 | 3.76 | SO 6675 1649 | *Quercus robur* |
| GLOR07 | 5.34 | SO 6679 1648 | *Quercus robur* |
| Dulwich Woods | N.D. | TQ 34340 72608 | *Quercus robur* or *Quercus petraea* |

Girth and National Grid Reference of the oak trees used in the present study, collected from the Horsepool Bottom Nature Reserve (GLOR01, GLOR02, GLOR03, GLOR07), or the Dulwich Woods. The girth of the core cross-section collected from the Dulwich Woods could not be determined (N.D.) as it was not possible to identify the position of the pith.

**Table 2. The dendrochronological analysis performed on the oak samples.**

| Tree ID | Sequence Date Range | Rings + unmeasured rings | Germination (circa, AD) | Storage |
|---|---|---|---|---|
| Dulwich Woods | n.d. | ca. 50 | ca. 1970 | RT |
| GLOR01 | AD1797-AD2013 | 216 | 1781 | RT |
| GLOR02 | AD1797-AD2013 | 216 | 1781 | RT, -20˚C |
| GLOR03 | AD1879-AD2013 | 135 + hollow | n.d. | RT |
| GLOR07 | AD1792-AD2013 | 221 | 1758 | RT, -20˚C |

Dendrochronological analysis performed on oak wood samples collected at Horsepool Bottom Nature Reserve (Gloucestershire, England) and used in the present study. GLOR07 and GLOR02 samples were collected in duplicates, a first replica was stored at room temperature (RT) while a second one was stored at -20˚C.

reporting the radial growth over time obtained from cores extracted from the same tree and dated (S1A Fig). Unless stated otherwise, the wood segments were decontaminated either by successive 5 min applications of a 5% bleach solution, a 60% EtOH solution, and a pure water bath, or by the mechanical removal of the segment surface with a scalpel, dried and homogenised to obtain wood powder. The wood powder was washed with TNE (200 mM Tris, 250 mM NaCl, 50 mM EDTA) and methanol (absolute methanol + 0.05 M CaCl2 + 40 mM DTT) buffers to reduce the content of phenolic compounds of the sample.

## DNA extraction and quantification

800 $\mu$l lysis buffer (Table 3) were added to each 2 ml tube containing 20–100 mg of wood powder. Tubes were incubated for 4 hours at 4°C or 65°C under 1,300 rpm constant agitation. At the end of the incubation; tubes were centrifuged for 5 minutes at 17,000 g and 4°C. The supernatant was recovered and transferred to a new 2 ml tube. 700 $\mu$l of chloroform were added to each tube, tubes were inverted 3 times and centrifuged at 17,000 g and 4°C for 5 minutes. The supernatants were retrieved and transferred to new 2 ml tubes. Two volumes of 99% ethanol were added to each tube. The tubes were inverted 20 times and stored at -20°C for 45 minutes. After the storage at -20°C, the tubes were centrifuged for 10 minutes at 17,000 g and 4°C; the supernatant was discarded. The precipitated DNA was resuspended in DEPC water (Sigma-Aldrich) and DNA concentration was measured by fluorometric quantification (Qubit 3.0 Fluorometer, dsDNA HS assay, ThermoFisher Scientific). In total, DNA was extracted from 427 samples obtained from 7 distinct oak wood cores/cross-section (S1 File).

## Heartwood DNA extract PCR inhibition assay

Amplification of chloroplast and nuclear DNA regions failed for Dulwich Woods, GLOR03 and GLOR07 heartwood while the amplification of chloroplast but not nuclear targets consistently succeeded for sapwood samples from the same trees (S1 and S2 Tables). Therefore, sapwood DNA was used as a positive control for the test of DNA amplification. To assess the presence of PCR inhibitors in heartwood DNA extracts, amplifiable Dulwich Woods sapwood DNA was titrated with oak heartwood DNA extract obtained from Dulwich Woods to reach the 1:4, 2:4, 3:4, 3.25:4 and 3.5:4 volume ratios (Fig 1). Sapwood-heartwood mix DNA was amplified using primers targeting a 203 bp region included in the rbcL (RuBisCO large subunit) chloroplast gene (S2 Table). Amplification products were observed after gel electrophoresis run.

**Table 3. DNA extraction buffers.**

|  | STE [26] | aDNA [24] | PTB [27] |
| --- | --- | --- | --- |
| SDS | 0.2/0.4/2% w/v | 2% w/v | 1% w/v |
| PVP | 6% w/v | - | - |
| NaCl | 100 mM | 10 mM | 5 mM |
| Tris-Cl pH 8 | 10 mM | 10 mM | 10 mM |
| EDTA | 1 mM | 2.5 mM | 10 mM |
| CaCl$_2$ | - | 5 mM | - |
| DTT | - | 40 mM | 50 mM |
| PTB | - | - | 2.5 mM |

DNA extraction buffers. SDS: sodium dodecyl sulfate; PVP: polyvinyl pyrrolidone (average molecular weight 40 kDa); EDTA: ethylendiaminetetracetic acid; DTT: 1,4-dithiothreitol; PTB: N-phenacylthiazolium bromide; Tris-Cl: Tris(hydroxymehtyl)aminomethane-chloride.

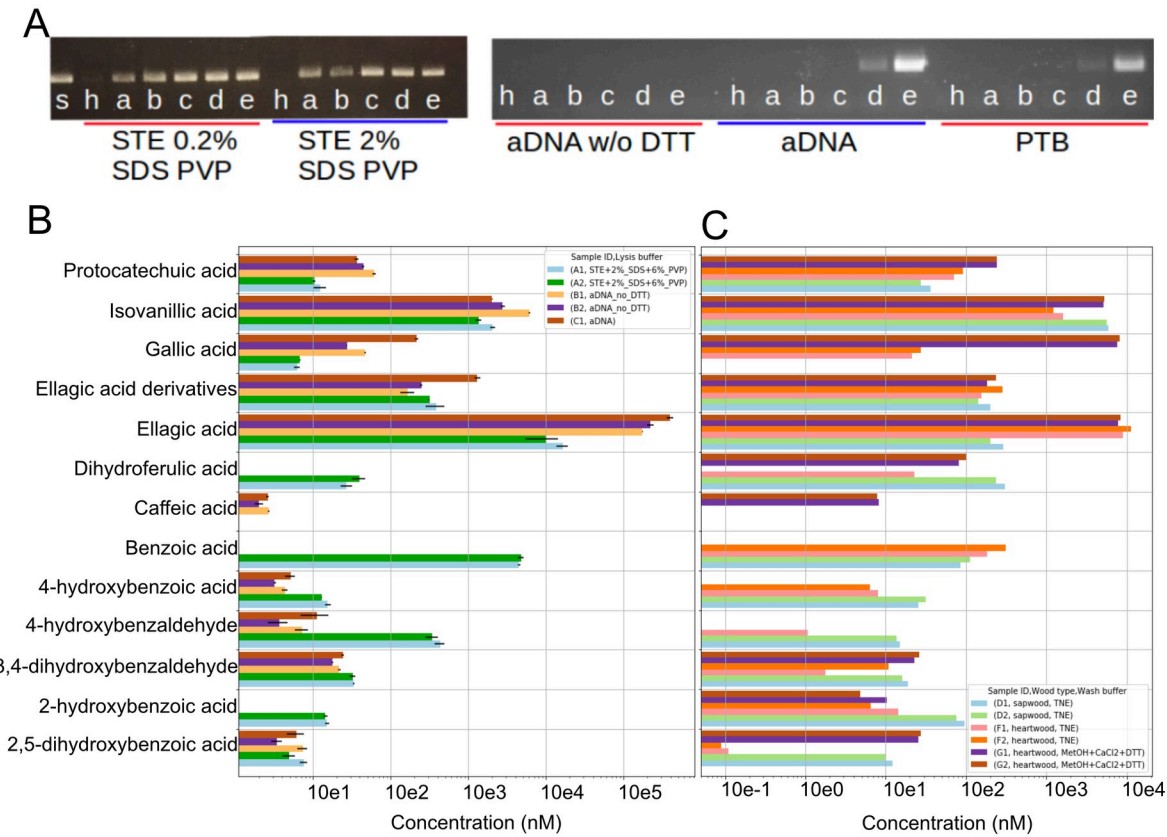

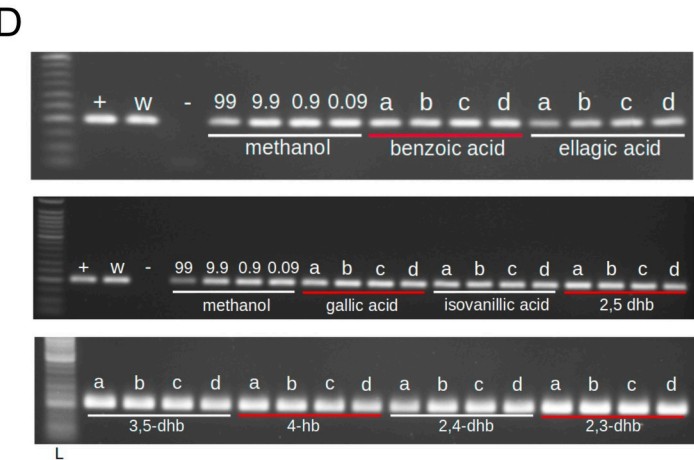

**Fig 1. PCR-inhibitory effect of heartwood DNA extracts.** A) PCR inhibitory effect of heartwood DNA extracts obtained with STE 2% SDS 6% PVP, STE 0.2% SDS 6% PVP, aDNA buffer with and without DTT, PTB buffer. Amplifiable sapwood DNA extract (s) was titrated with heartwood DNA extract (h) at 1:4 (a), 2:4 (b), 3:4 (c), 3.25:4 (d), 3.5:4 (e). Sapwood-heartwood extract DNA was amplified by PCR for 40 cycles. The two panels of this figure were taken from different gels under the same experimental conditions. B, C) Concentration of phenolic compounds and ellagic acid derivatives in DNA extracts obtained from oak heartwood (B) and from the washing surnatants of oak sapwood and heartwood (C). Error bars represent the standard deviation. D) Concentration of the tested compound: a) 1.0 mM, b) 0.1 mM, c) 0.01 mM, d) 0.001 mM. Sapwood (+), 50% diluted sapwood (w), PCR negative control (-). Hb: hydroxybenzoic acid, dhb: dihydroxybenzoic acid. Ellagic acid is dissolved in methanol which is why controls with the addition of only methanol to the PCR were carried out (99%, 9.9%, 0.9%, 0.09%). The upper, middle, and lower panels of this figure were taken from different gels under the same experimental conditions.

## Quantification of phenolic compounds in wash buffer supernatants and DNA extracts

To identify the heartwood compounds possibly responsible for PCR inhibition, phenolic compounds were quantified in the DNA extracts, TNE and methanol/CaCl$_2$/DTT washing supernatants by ultra-high-performance liquid chromatography-quadrupole time-of-flight mass spectrometry (UPLC QTOF-MS) using a validated method [28, 29]. We also estimated the concentration of total phenolic compounds in DNA extracts using a protocol based on the Folin–Ciocalteu (F–C) reagent [30].

## Phenols PCR inhibition assay

PCR-amplifiable sapwood DNA was mixed with analytical standard of the phenolic compounds detected by UPLC QTOF-MS of buffer supernatants and DNA extracts to test their PCR inhibition potential. 2 mM, 0.2 mM, 0.02 mM and 0.002 mM concentrations in the PCR mix were tested for each phenolic compound. We included a methanol control as the phenolic acid powder was dissolved in 99% methanol.

## Molecular dynamics and metadynamics simulations

The NAMD 2.9 molecular dynamics package [31] was used to investigate the interaction of ellagic acid with a 20-nucleobase double-stranded DNA (dsDNA) model in water and in methanol, with the goal of providing atomistic cues able to aid the purification of dsDNA from polyphenol contaminants and potentially of rationalising how this phenolic compound may inhibit PCR. We generated a 20-nucleobase dsDNA model with the sequence CTGGGA-CATGGACAACTGTG, which corresponds to one of the rbcL-targeting PCR primers. We numbered the nucleobases in this sequence from 1 to 20 and the complementary ones from 40 to 21, as shown in Fig 2A. We indicated minor-groove sites by lower case Roman numbers and major-groove by capital Roman numbers. Specifically, minor-groove site *i* was identified with respect to the phosphorus atoms of bases 5 and 40, *ii* of bases 6 and 39, ..., *xvi* of bases 20 and 25, while we identified major-groove site *I* with respect to the phosphorus atoms of bases 2 and 34, *II* of bases 3 and 33, ..., *XIII* of 14 and 22 (Fig 2A). We added 38 Na$^+$ counterions and a 10 Å solvent buffer of either water or methanol in a periodically repeated orthorombic supercell.

We set a time step of 2 fs and we constrained the length of the bonds containing hydrogen with the SHAKE algorithm [32]; the Particle Mesh Ewald method was used for the electrostatic interactions and a cut-off of 12.0 Å for the non-bonded interactions. We kept the temperature at 300 K by a Langevin thermostat [31] with a collision frequency of 1.0 ps$^{-1}$; we maintained pressure at 1 atm by a Langevin piston [33] with a 200 fs oscillation period and a 100 fs damping time constant. To avoid fraying and DNA crossing the periodic boundaries, the dsDNA model had the phosphorous atoms of its terminal bases on each side restrained with harmonic potentials of spring constant equal to 2.5 kcal/mol/Å$^2$.

We used the Amber force-field with parmbsc1, a parametrization developed and optimized specifically for nucleic acids [34] and the TIP3P model for water. We parametrized ellagic acid with the General Amber Force-Field (GAFF) for organic molecules [35]. We optimised the structures of nine ellagic acid conformers, characterised by different intra-molecular hydrogen bond patterns, at the density functional theory level with the B3LYP exchange and correlation functional and a 6–31G* basis set with Gaussian09E [36]. We evaluated partial charges by means of a multiconformational RESP fitting [37, 38] at the Hartree Fock level, for consistency with the Amber force-field parametrization. To prevent $\pi$-$\pi$ stacking of ellagic acids, we

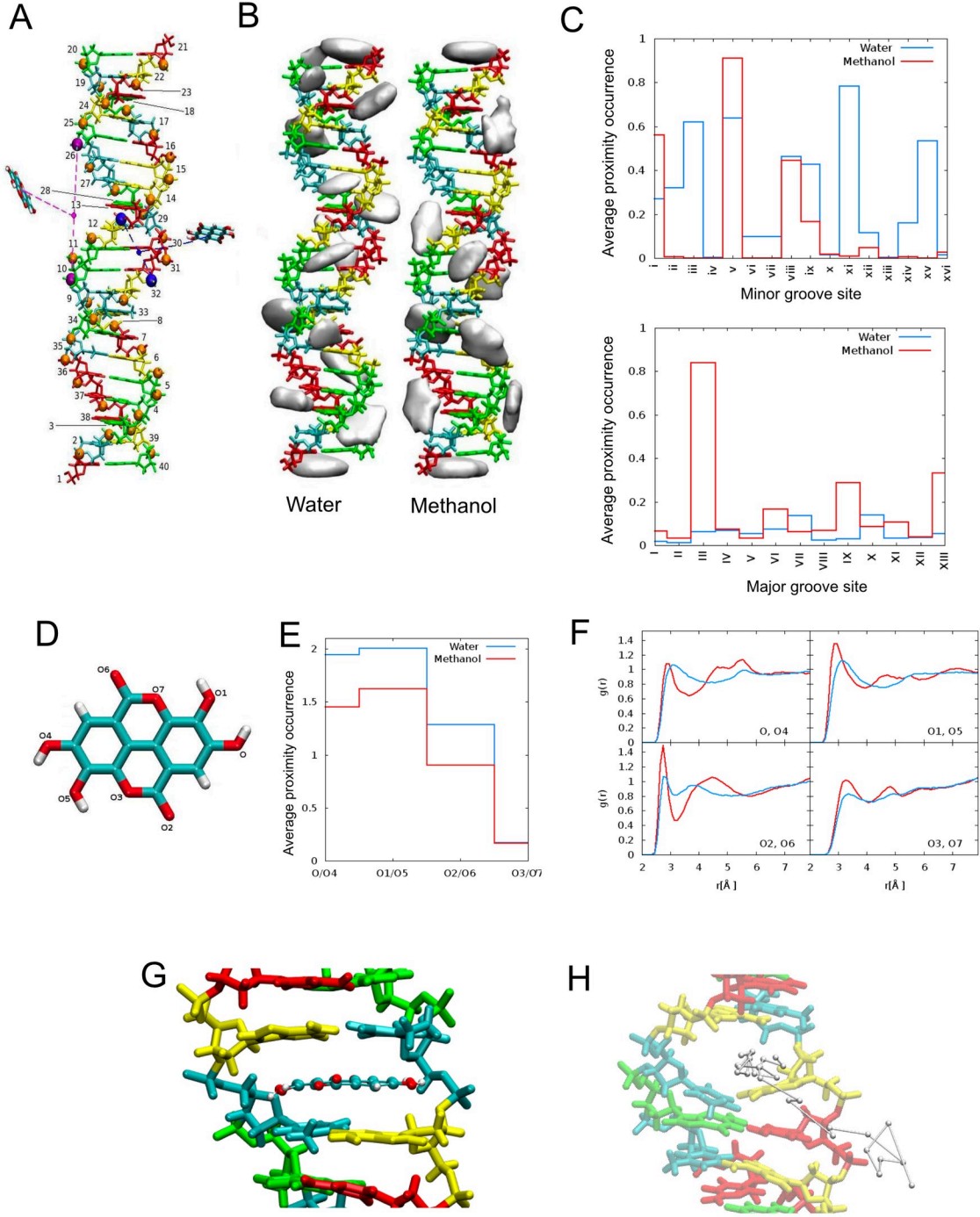

**Fig 2. Interactions between ellagic acid, solvents and DNA.** A) dsDNA model with labels (1 to 40) for the nucleobases. Cytosine is in red, adenine in yellow, thymine in cyan, guanine in green, and phosphorus atoms in orange. An exemplar minor-groove site (viii) is shown in blue and an exemplar major-groove site (IX) is shown in purple, with the relevant distances for the proximity calculation of ellagic acid. B) Density maps of ellagic acid atoms in interaction with dsDNA in water and in methanol. The isosurfaces correspond to a density of 0.5 Å$^{-3}$. C) Average proximity occurrences in the minor groove sites (top) and in the major groove sites (bottom). D) Ellagic acid with labelled oxygens. E) Average number of hydrogen bonds between ellagic acid oxygens and water (blue) and methanol (red) oxygens, averaged for each pair of equivalent ellagic acid oxygens. F) Pair correlation function ($g(r)$) of ellagic acid oxygens and water (blue) and methanol (red) oxygens, averaged for each pair of equivalent ellagic acid oxygens as labelled in the ellagic acid model. G) A representative molecular dynamics snapshot of the intercalated ellagic acid within DNA bases. H) An example of ellagic acid unbinding trajectory obtained with metadynamics. This image was made with VMD and is owned by the Theoretical and Computational Biophysics Group, NIH Center for Macromolecular Modeling and Bioinformatics, at the Beckman Institute, the University of Illinois at Urbana-Champaign (www.ks.uiuc.edu).

modified the parameters of the aromatic ring carbons and oxygens involved in inter-ellagic acid van der Waals interactions to $\epsilon_i = 1.2 \times 10^{-3}$ kcal/mol and $\sigma_i = 6.5 \times 10^{-1}$ Å; this modification did not affect the interaction with DNA and/or the solvent [39]. To identify preferential interaction sites and enhance the sampling, we carried out production MD NPT runs for statistics collection for 80 ns for ellagic acid in water or methanol, for 500 ns for the dsDNA model in a concentrated ($\sim$ 0.18 mol/L) ellagic acid/water or methanol solutions (containing 40 ellagic acid molecules), and for 100 ns for a representative intercalated system as a preparation for metadynamics simulations. We obtained the latter with a docking procedure using AutoDock 4.2 [40].

To qualitatively explore the unbinding of the intercalated ellagic acid which would not occur on typical MD time scales, simulations were carried out with the enhanced sampling method metadynamics and the PLUMED 2.3 plugin [41–43], biasing the distance between the centre of mass of the intercalated ellagic acid and the centre of mass of the aromatic rings of the four nucleobases of the intercalation site. We deposited Gaussians, with a 0.085 Å width and a 0.1 kcal/mol height every ps. For each solvent, we performed eight metadynamics simulations, starting from different initial positions along the previous MD trajectory, and stopped when unbinding occurred. We performed the analysis by means of the CPPTRAJ [44] and MDAnalysis software [45, 46].

We defined hydrogen bonds as having the donor-acceptor distance smaller than 3.5 Å and the donor-H-acceptor angle larger than 120˚. We defined proximity of ellagic acid into the minor or major grooves to occur when the distance of the ellagic acid centre of mass from the centre of mass of the two phosphorous atoms on opposite sides of the groove (Fig 2A) was smaller than 3 Å (roughly equal to the minor semi-axis of ellagic acid). We obtained density maps of ellagic acid to identify interaction hotspots by modelling each of its atom as a normalized Gaussian of width equal to the atomic radius, and averaging the distribution over the MD NPT production runs.

## Preparation of WGS and WGBS libraries

WGS and WGBS libraries were prepared following a double-stranded DNA blunt-end adapter ligation protocol [47] from extracts containing 1 to 10 ng of DNA. WGBS libraries were prepared using methylated Illumina-compatible adapters with overhangs (Eurofins), that is, adapters with the same sequence as above where each cytosine is methylated at the 5-carbon position (5mC). Filling-in of the adapter overhangs took place in the presence of dATP, d5mCTP (Zymo Research), dTTP and dGTP (25 mM each). After the adapter fill-in step, DNA libraries were converted using the EZ DNA methylation Gold kit (Zymo Research). Indexing-PCR was performed for 24 cycles for both WGS and WGBS libraries with a proof-reading but uracil-tolerant DNA polymerase (Phusion U Hot Start DNA polymerase, Thermo Scientific, cat. no. F555S), and adapter-dimers were removed as for WGS libraries [47]. The WGS library preparation protocols were performed on DNA extracts obtained as reported in S3 Table (column named DNA extraction buffer) considering a minimum of three technical replicas per each sample. The success of the library preparation was established by the presence of a continuous smear, corresponding to the adapter-ligated and amplified DNA fragments, in the electrophoresis gel in which the products of the indexing PCR were run. Libraries were sequenced by a MiSeq Reagent Kit v3 75 bp paired-end chemistry.

## Analysis of WGS and WGBS data

Adapter sequences at the 3' ends of reads and reads having a Phred score below 20 were removed by TrimGalore [48]. Filtered reads were mapped to the oak (*Quercus robur*) reference

genome [49] with the addition of the Enterobacteria phage phiX174 genome (NCBI reference sequence: NC_01422.1) as a separate contig using Bowtie2 [50]. PCR-duplicates were removed by the samtools command rmdup and deduplicated DNA reads were used for the downstream analysis [51]. Genome coverage was calculated using the bedtools command coverage [52]. We assessed the damage pattern of DNA reads by mapDamage under default setting [53]. WGBS reads were mapped to the oak reference genome [49] and cytosine methylation levels were quantified using bs-seeker2 with 4 mismatches allowed [54]. Bisulfite-conversion efficiency was calculated as the mean percentage methylation value of cytosines in the chloroplast genome, which is un-methylated [55]. The presence of potential biases introduced by bisulfite treatment or the subsequent PCR was assessed by the bismark package bam2nuc [56]. The methylation level of 100 bp nuclear genome windows having at least 5x coverage was compared in sapwood and heartwood samples of the same tree. The 5x coverage threshold was chosen for being higher or equal than the coverage value adopted in plant epigenomics studies based on WGBS [57–59]. The DNA methylation median variation between sapwood and heartwood was calculated in each tree for each of the informative 100 bp regions. The software Preseq was used to assess the complexity of the WGS libraries and the gain in unique DNA sequences potentially obtainable by additional sequencing of the same libraries [60].

## Results

### Heartwood DNA extracts contain PCR inhibitors

Sample preparation and DNA extraction protocols were optimised with the aim of maximizing the amount of retrieved DNA and reduce contaminants that co-precipitated with the DNA. Sample homogenisation and washing significantly affected DNA yield (S1B and S1C Fig). Similarly, lysis buffer composition was critical to maximise the amount of DNA extracted from wood (S1D Fig).

Following DNA extraction optimisation, we attempted to validate the origin of the DNA extracted from oak sapwood and heartwood by means of PCR with oak-specific primers (S1 and S2 Tables). We repeatedly failed in the amplification of DNA extracted from heartwood samples so we sought to determine the inhibitory effect of heartwood DNA extracts by an end-point PCR experiment (40 cycles) using primers that targeted a 203 bp region included in the chloroplast gene rbcL (Ribulose bisphosphate carboxylase large chain) (S1 and S2 Tables). A single sapwood DNA extract was titrated with heartwood DNA extracts obtained with distinct lysis buffers at increasing quantitative ratios (Fig 1A). Consistently, amplification succeeded for the pure sapwood but not for the pure heartwood DNA extract controls (Fig 1A). The addition of heartwood DNA extracted with the STE 0.2/2% SDS, 6% PVP buffer did not impair the amplification of sapwood DNA. In contrast, the heartwood DNA extracts obtained with the ancient DNA (aDNA) without DTT buffer inhibited the PCR in all of the tested mixes while heartwood DNA extracts obtained with the aDNA and N-phenacylthiazolium bromide (PTB) buffers impeded the amplification of sapwood DNA at 1 in 4 (1:4), 2:4 and 3:4 quantitative ratios (Fig 1A, Table 3 for lysis buffer composition). Heartwood DNA extracts might have contained oak DNA but that DNA was not amplifiable and thus, was unlikely to have contributed to the successful amplification reactions. At the same time, the sapwood DNA extract could have contained PCR inhibitors but their amount was insufficient to prevent amplification from pure sapwood DNA extract. Therefore, the inhibition of DNA amplification observed for part of the sapwood-heartwood mixes must be attributed to the inhibitors that were present in the heartwood extracts.

To identify specific, potentially PCR-inhibiting phenolic compounds in oak heartwood DNA extracts, we determined the concentrations of phenolic compounds and ellagitannins by

UPLC QTOF-MS in PCR-inhibiting extracts obtained with the PTB and aDNA (with and without DTT) buffers and in non-inhibiting extracts obtained with the STE 2% SDS 6% PVP buffer (Fig 1B). We focused on phenolic compounds as those are abundant in heartwood and some have been reported to inhibit PCR [9, 61]. The heartwood DNA extracts obtained with the PCR-inhibiting aDNA buffers contained 15-fold more ellagic acid than those obtained with the STE buffer, by far the largest difference across the twelve quantified phenolic compounds. In absolute terms, ellagic acid also was by far the most abundant of the quantified phenolic compounds, reaching a concentration of 400 $\mu$M (Fig 1B). We also analysed by UPLC QTOF-MS the phenolic compounds and ellagitannins present in the washing buffers that were used during the preparation of the samples for the DNA extraction, with the aim of identifying the compounds that were most effectively removed during the washing of the wood powder (Fig 1C). Overall, the methanol buffer retains more of the tested phenolic compounds but both TNE and methanol buffers were used to maximise the removal of phenolic compounds [62, 63]. To directly test the inhibitory effect of specific phenolic compounds, we added the respective analytical standard used for the UPLC QTOF-MS quantification to PCR reactions known to successfully amplify rbcL from sapwood DNA extract. Each phenolic compound standard was added at a final concentration of either 0.001, 0.01, 0.1 or 1.0 mM. In contrast to the above results with added aDNA buffer-extracted heartwood DNA (Fig 1A), we did not observe PCR inhibition with the addition of any of the tested pure phenolic compounds at any of the tested concentrations (Fig 1D). Even though a very high concentration of ellagic acid is characteristic of heartwood DNA extracts obtained with the aDNA buffer, ellagic acid is therefore unlikely to be responsible for the PCR inhibition caused by aDNA buffer-extracted heartwood DNA.

## Ellagic acid binds to DNA in a solvent-specific manner

Linear dichroism spectroscopy experiments had previously shown that ellagic acid binds to double stranded calf thymus DNA, inclined at an angle between 66˚ and 90˚ with respect to the DNA axis, proving that ellagic acid is able to intercalate within DNA [64]. Given its high concentration in our DNA extracts and the evidence reported in the literature, ellagic acid was selected as the representative phenolic compound in order to study its interactions with a double-stranded DNA model by means of MD simulations in either water or methanol. We used molecular dynamics to elucidate the interaction mechanisms between phenolic compounds, solvent and biomolecules at the atomistic level [65, 66]. Simulations in water mimicked the PCR reaction environment, while simulations in methanol aimed at elucidating the mechanism associated with increased amplification efficiency for DNA extracted from methanol-washed samples [14]. Methanol has a reduced capacity to form hydrogen bonds with respect to water and is a bulkier molecule. This was already reflected in the MD simulations of ellagic acid in the two solvents in absence of dsDNA: in fact ellagic acid formed on average more hydrogen bonds with water (10.8) than with methanol (8.3), with the reduction affecting the hydrogen bonds involving all oxygens except the intra-ring ones, as shown in Fig 2D and 2E. The pair correlation functions ($g(r)$, Fig 2F) of each ellagic acid oxygen with the solvent oxygen atoms showed sharper first neighbour peaks in methanol than in water, suggesting a more rigid and structured first neighbour solvation shell around ellagic acid in methanol. MD simulations of the dsDNA model with multiple ellagic acid molecules revealed that the phenolic compounds interacted with dsDNA by means of hydrogen bonds with the phosphate groups in the major grooves and van der Waals interactions in the minor grooves. Favourable interaction hotspots were identified through the density maps of ellagic acid atoms in Fig 2B, which showed an increase in favourable interaction sites in water relative to methanol. Using the ellagic acid to DNA proximity criterium previously defined, we identified a larger average

number of ellagic acid molecules close and therefore interacting with dsDNA in major grooves in methanol (2.2) relative to water (0.7), while the opposite is true for minor grooves (2.2 in methanol, 4.6 in water). Overall, interactions within the grooves were less frequent in methanol than in water. Furthermore, the overall average number of hydrogen bonds between dsDNA and all the polyphenols in all locations (within and outside the grooves) was 50.3 in water and 46.2 in methanol (Fig 2C). The average dsDNA phosphorus-phosphorus distance in the minor grooves was smaller in water (10.9 ± 1.5 Å) than methanol (12.2 ± 1.6 Å); conversely, the average phosphorus-phosphorus distance in the major grooves was larger in water (20.3 ± 1.8 Å) with respect to methanol (18.1 ± 1.7 Å). This favoured in methanol a configuration of ellagic acid pinned by two hydrogen bonds at opposite sides of the major grooves, while in water, due to the larger major groove space, the hydrogen bond interaction was on either side resulting in an increased freedom to move in and out the major grooves. These data suggest that ellagic acid on average tends to stay closer to and form more interactions with dsDNA in water than in methanol.

While the MD simulations with multiple ellagic acids provided indication on the sites of approach and attachment of ellagic acid to dsDNA, because of time scale limitations intrinsic to this technique, they do not allow us to directly observe the activated process of intercalation from the solvent, which would require deformation of the DNA structure to accommodate the phenolic compound. We modelled the intercalation of ellagic acid between dsDNA nucleobases [8(A)-33(T);9(T)-32(A)] as a representative case. The intercalated pose was stabilized by $\pi$-$\pi$ stacking interactions between the aromatic rings of ellagic acid and those of the lower and upper bases, that expanded the space in between them to accommodate the phenolic compound, from an average distance of 3.5 ± 0.6 Å to 6.7 ± 0.2 Å in water. Starting from the stable intercalated ellagic acid at the end of the MD in water, the solvent was removed and substituted by methanol. In the following production NPT MD, ellagic acid remained intercalated, with an average distance in between the upper and lower bases of 6.8 ± 0.2 Å, very similar to that in water, showing that intercalated structures can also be stable in methanol (Fig 2G).

Metadynamics simulations provided indications of favourable unbinding paths. In all simulations starting from different initial conditions, ellagic acid exited the intercalation site from the major groove, as shown in Fig 2H for a typical trajectory, with a similar number of steps in both water and methanol.

## WGS library preparation success is limited to oak heartwood samples less than 100 years old

The failure of heartwood DNA amplification prevented us from validating the presence of oak endogenous DNA in this tissue. Therefore, we attempted to reach this aim by WGS, successfully used for the characterisation of archaeological samples, including heartwood, that contain a limited amount of DNA and present contaminants potentially acting as enzyme inhibitors [24, 67]. We prepared WGS libraries from wood samples spanning the AD 1800–2014 (0–214 years old) time period to test DNA library preparation success over wood age (Fig 3A). The success of the WGS library preparation process was assessed by checking for the presence of a continuous smear visible in the electrophoresis gel as reported in the reference protocol [47].

We extracted DNA using an STE-PVP buffer as it did not inhibit PCR with the addition of PTB to break the interactions between DNA and proteins [68]. We prepared libraries from samples that were treated considering one of the two storage, homogenisation and washing variants reported in S3 Table. WGS library preparation usually was unsuccessful for samples older than 80 years old, except for GLOR07 fresh-frozen samples from which libraries could successfully be prepared up to an age of ∼ 100 years and for some exceptionally old (older

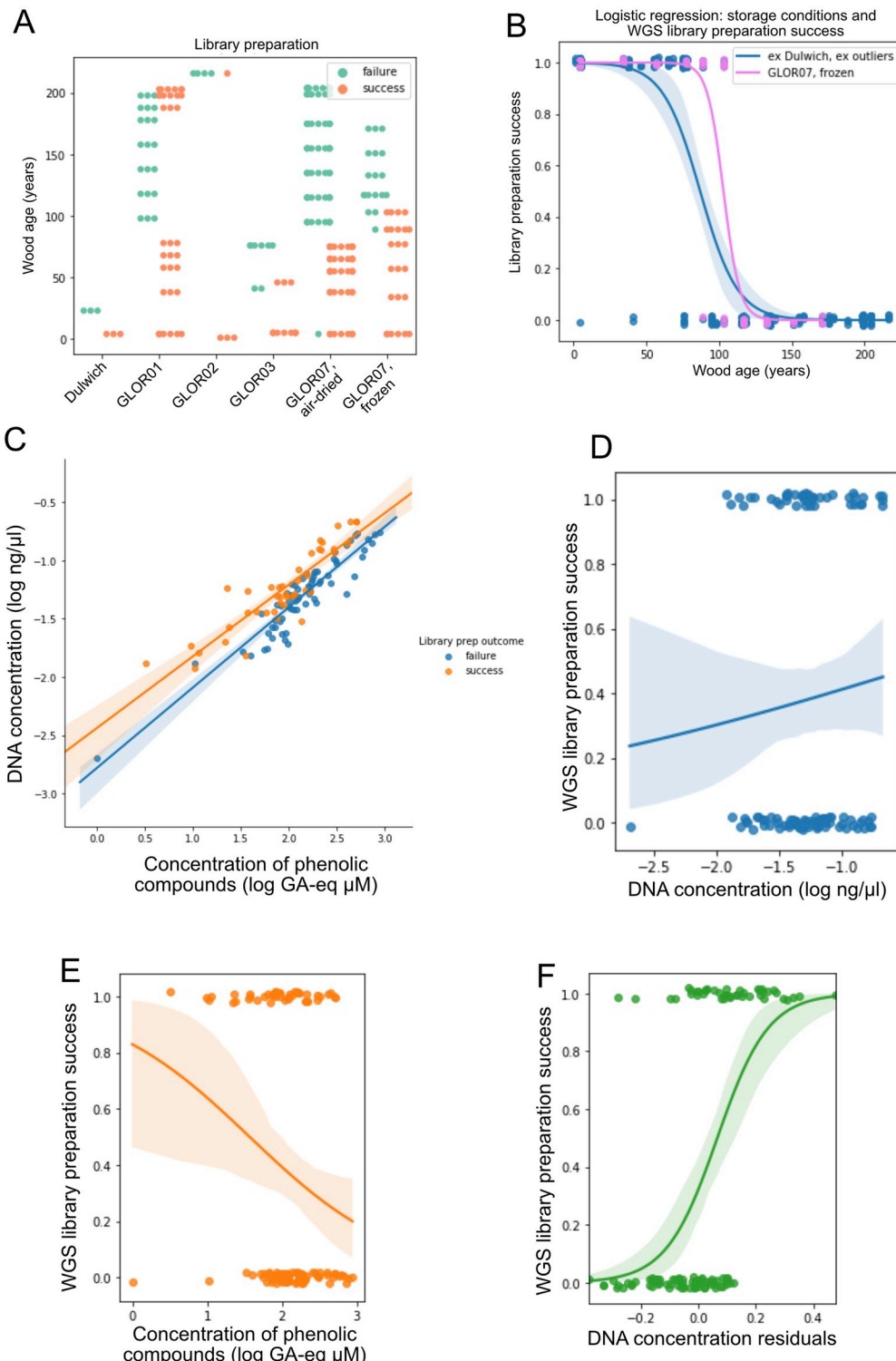

**Fig 3. Analysis of the factors affecting the success of DNA library preparation.** A) Success of WGS library preparation performed from Dulwich Woods and GLOR01–07 trees (Table 2). Two distinct samples were collected from GLOR07, one was air-dried and stored at room temperature while the other was frozen and stored at -20˚C. B) Logistic regression of the relationship between the age of wood (years) and the WGS library preparation success in GLOR07 frozen samples ("GLOR07, frozen") and the GLOR01-GLOR07 samples that were air-dried and stored at room temperature ("ex Dulwich, ex outliers"). The logistic regression was performed after excluding ("ex") the samples

obtained from the Dulwich Woods ("Dulwich") core and those that contained a low amount of oak DNA ("outliers"). C) WGS library preparation outcome of samples correlated with the concentration of phenolic compounds (log of gallic acid equivalents $\mu$M) and the concentration of DNA (log ng/$\mu$l). D, E, F) Logistic regression relating the success of WGS library preparation to D) the concentration DNA as measured by fluorometric quantification (log ng/$\mu$l), E) the concentration of phenolic compounds (log of Gallic acid equivalents $\mu$M), F) the residuals of DNA concentration.

than 180 years) GLOR01 and GLOR02 samples, but the sequencing of three of these libraries revealed that the contained DNA was not oak DNA (S3 Table), which we assume to also be true for the libraries that were not sequenced. The overall comparison between the storage conditions "air-dried, silica" and "frozen" was also accomplished by the logistic regression of the WGS library outcome as a function of the storage conditions (Fig 3B). The age-dependence of WGS library preparation success was significant (logistic regression P(z<6.443)<0.0005) (Fig 3B).

We also assessed how the concentration of DNA and phenolic compounds could have limited the success of WGS library preparation. We first stratified the samples used for WGS library preparation on the basis of their DNA and phenolic compounds concentration (Fig 3C). We observed that successful WGS libraries were obtained from DNA extracts having higher concentration of DNA (Fig 3C and 3D). We also analysed the relationship between the content of phenolic compounds and success of WGS library preparation (Fig 3E). The content of phenolic compounds negatively correlated with the success of WGS library preparation and predicts the WGS library preparation success better than the concentration of DNA (Fig 3E). The prediction can be further improved by performing a background correction of the DNA concentration measurements by the content of phenolic compounds (DNA concentration residuals, Fig 3F). By doing that, it was possible to conclude that WGS library preparation had an 80% chance of success when the concentration of DNA residual was greater than 0.2, that is when the DNA concentration was 1.6-fold greater than the modelled background.

## Oak heartwood DNA molecules present an age-dependent fragmentation pattern

Oak heartwood libraries included in the AD 1902–1980 and AD 1810–1820 time periods and the corresponding sapwood samples (AD 1990–2014) of four trees were sequenced generating 902,770 to 24,761,174 DNA reads per sample. PCR duplicates-removed reads mapped to the oak genome ranged from 43% to 75% of the sequenced reads for the 1902–2014 samples providing evidence for the presence of a remarkable amount of endogenous oak DNA in heartwood (S3 Table). Only 0.14%-0.72% of reads obtained from the ~200 years old heartwood samples (AD 1811–1815, 1816–1820, 1821–1830) had oak origin, leading us to exclude these samples from the downstream analysis (S3 Table). AD 1811–1830 DNA libraries were obtained from a wood core (GLOR01) decontaminated by removal of the external layer of wood rather than bleach, ethanol and water washing to test for the effect of the decontamination protocol on the library preparation success. Considering that the vast majority of DNA contained in these libraries derived from contamination, we suggested decontamination by washing is preferable over removal of wood core surface. At the same time, we no longer considered sample decontamination as a critical factor for library preparation success leaving sample storage (room temperature vs -20˚C) as the only step that limited the age of heartwood samples from which a library can be obtained. In particular, the storage at -20˚C extended to ~100 the years of age of the oldest heartwood samples from which it was possible to obtain a WGS library having a sufficient amount of oak endogenous DNA (sample GLOR07 1902–1915, Deduplicated, nuclear-mapped reads = 51.86% of the sequenced reads, S3 Table).

Nucleic acids are degraded and cell structure is lost during sapwood-to-heartwood transi-tion thus exposing the residual DNA molecules to the environment present in the tree trunk which might cause its further degradation [2]. We assessed the length of nuclear DNA frag-ments and DNA damage to possibly identify patterns of DNA decay over sapwood-to-heart-wood transition and heartwood ageing [53]. Heartwood DNA fragments are shorter than sapwood's in all the considered trees. DNA fragment length further decreased between GLOR07 AD 1964–1980, 1934–1944 and 1902–1915 heartwood (Fig 4A–4D), suggesting that the degradation of DNA continues after the formation of heartwood.

We also investigated the mechanism by which DNA fragmentation might have occurred by analysing the pattern of nucleotide composition of oak heartwood DNA. In aDNA, post-mor-tem DNA fragmentation can occur by hydrolysis of phosphodiester bond or by elimination of a purine nucleotide [69]. Fragmentation generates over-hanging DNA molecules in which deam-ination is expected to increase exponentially toward the damaged DNA fragment end [70]. We quantified the sapwood and heartwood samples cytosine-to-uracil (sequenced as thymine) deamination pattern and composition at genomic position preceding read start and following read end using MapDamage [53]. None of the sequenced samples reported deamination at read ends or purine enrichment in the genomic regions surrounding the reads in contrast with the pattern observed for authenticated oak aDNA samples [24] (Fig 4E and 4F).

## DNA degradation by sodium bisulfite impedes the profiling of oak heartwood DNA methylation

The high content of endogenous nuclear DNA contained in some WGS libraries led us to pre-pare and sequence WGBS libraries from a subset of the same sapwood and heartwood samples. The selected samples were GLOR03 sapwood 1980–2013, GLOR03 heartwood 1962–1972, GLOR07 sapwood 1990–2014, GLOR07 heartwood 1964–1980. WGBS libraries were prepared using a modified version of a double-stranded DNA (dsDNA) blunt-end adapter ligation pro-tocol [47] in which standard adapters were replaced by cytosine-methylated adapters and in which bisulfite treatment was performed after the adapters fill-in step. We analysed WGBS reads to profile DNA methylation at genome-wide level. We identified 178 (GLOR07) and 66 (GLOR03) 100 bp regions, having at least 5x coverage in sapwood and heartwood samples from the same tree core and thus informative for DNA methylation analysis. 56 of these regions were common to both trees. Such a low number of genomic regions covered by a sufficient number of reads impeded a proper analysis of DNA methylation profile at genome-wide level. By comparing WGS and WGBS sequencing data, we also observed a decrease in the fraction of genomic bases covered by at least one DNA read (Fig 5A). The only differences between the WGS and WGBS protocols were the replacement of standard sequencing adapters with cyto-sine-methylated adapters, and treatment of the adpter-ligated DNA fragments with the sodium bisulfite. The use of cytosine-methylated adapters is unlikely to have compromised the outcome of the sequencing, while sodium bisulfite is known to being able to degrade DNA [71]. There-fore, the treatment of DNA by sodium bisulfite reduced the diversity of the DNA reads that were sequenced and impeded the profiling of DNA methylation at genome-wide level.

The WGS and WGBS libraries were sequenced using an Illumina MiSeq platform which is the one generating the lowest number of reads per run among the Illumina platforms. Despite this shortcoming, we chose the MiSeq because, at first, we wanted to confirm the presence of endogenous nuclear DNA in our set of samples, an aim that can be reached by the MiSeq capacity. In addition, performing a sequencing run with higher-throughput sequencers is justi-fied only with WGS libraries having a sufficient complexity. We analysed the DNA reads gen-erated by the sequencing of oak sapwood and heartwood samples using the software Preseq to

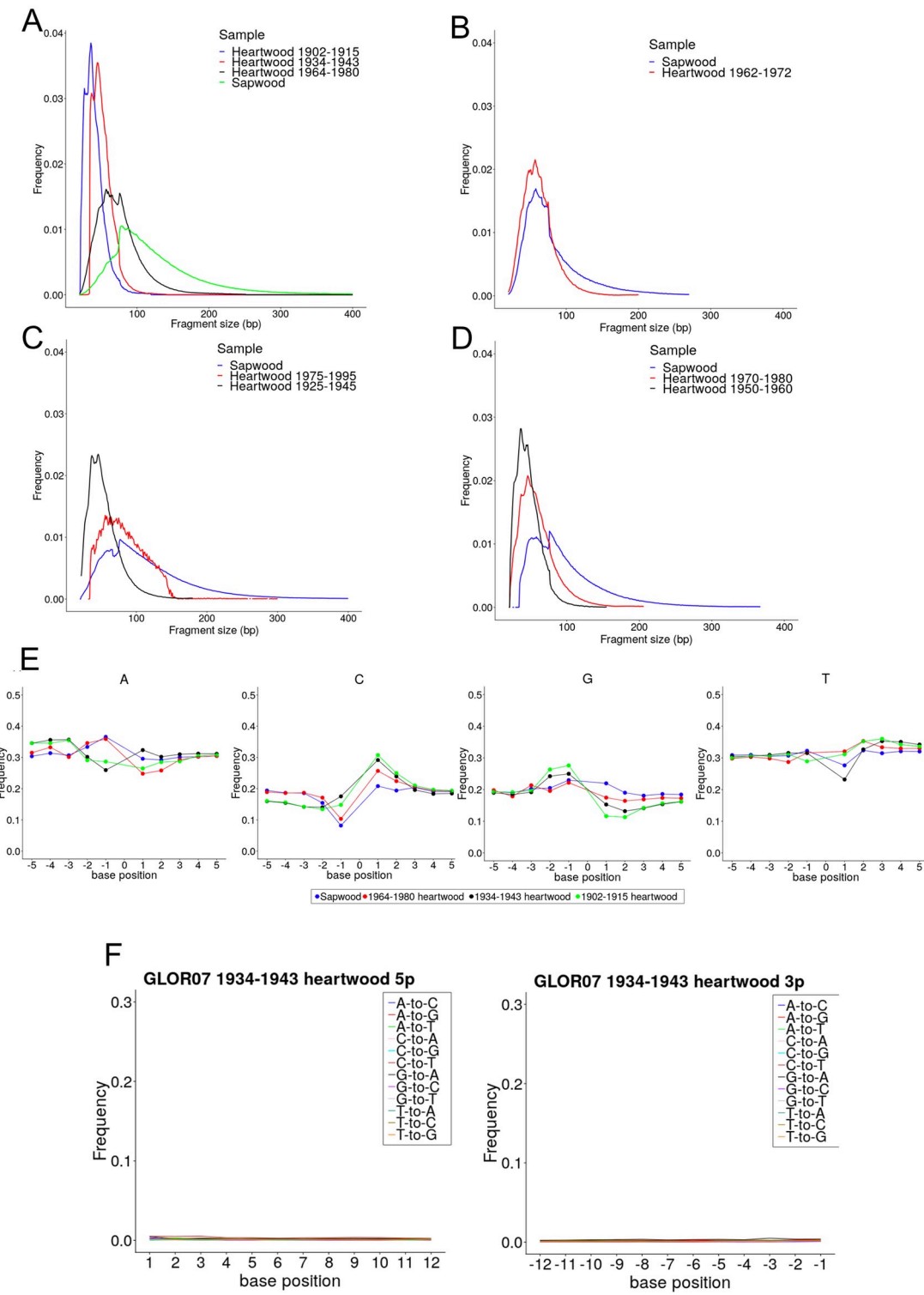

**Fig 4. DNA fragmentation and damage pattern in oak heartwood.** A-D) DNA fragments length distribution profile of (A) GLOR07, (B) GLOR03, (C) GLOR02 and (D) GLOR01 sapwood and heartwood DNA fragment length profile. We reported the fraction of total DNA fragments corresponding to each length (bp). E) Nucleotide frequency at the 3' (-1 to -5) and 5' (1 to 5) 5bp genomic positions of DNA reads in the core sample GLOR07. F) Misincorporation frequency at DNA read 5'- and 3'-end in the core sample GLOR07 1934–1943.

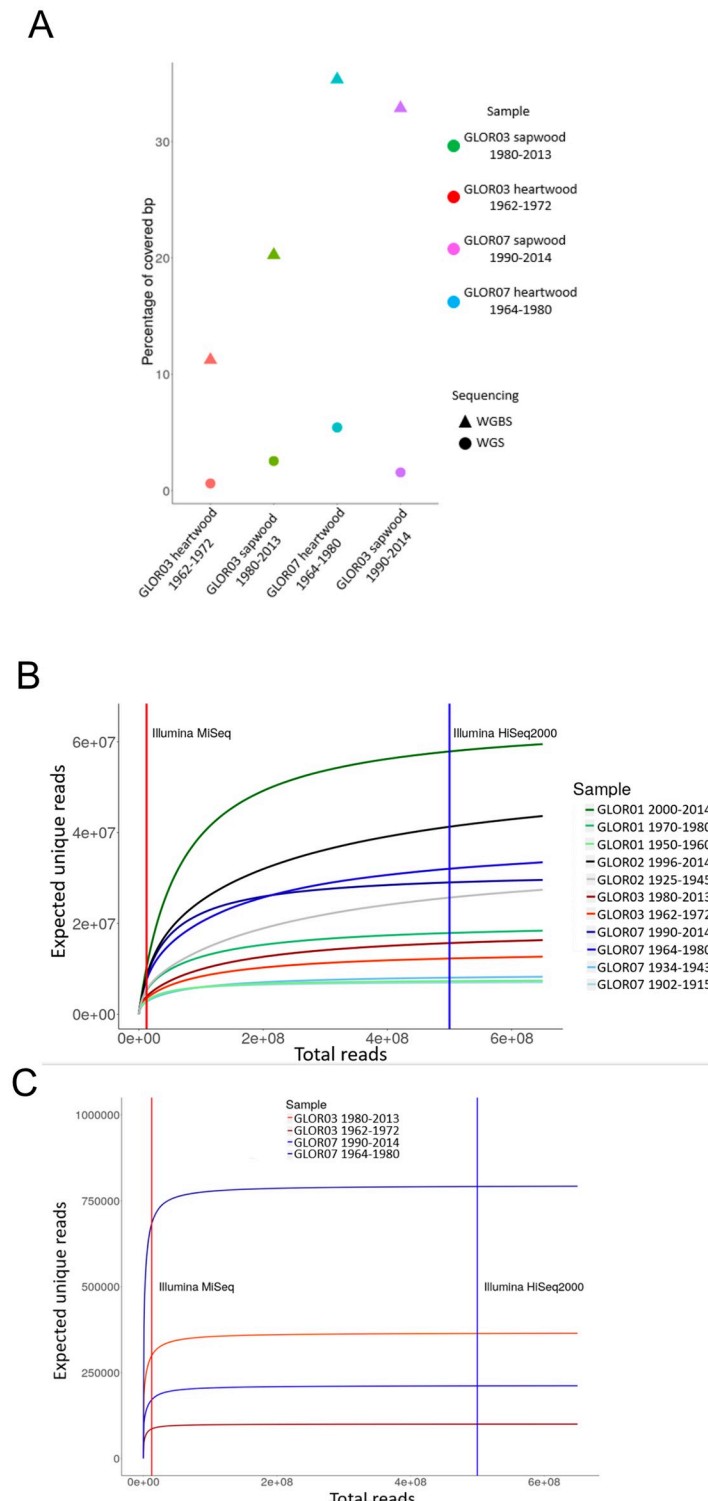

**Fig 5. WGBS is not feasible in oak heartwood.** A) Percentage of genomic bases covered by at least one WGS or WGBS DNA read generated from the sequencing of the GLOR07 and GLOR03 sapwood and heartwood samples. B-C) Number of distinct reads expected to be obtained at the increase of the total number of reads as predicted by Preseq [60]. We included the WGS (B) and WGBS (C) libraries having a number of endogenous reads compatible with the software requirement. We reported the number of reads generated by an Illumina MiSeq (21 hours run, chemistry version 3, 75 bp reads) and Illumina HiSeq2000 (8 days run, 100 bp reads).

assess the complexity of the sequencing library and evaluate if further sequencing experiments at higher throughput would have resulted in increasing the fraction of the genome covered by the DNA reads [60].

The prediction indicated that all of the WGS libraries would have generated further distinct reads if sequenced with platform having a throughput higher than the Illumina MiSeq (Fig 5B). Therefore, further sequencing of the WGS libraries would have increased the fraction of the genome covered by the DNA reads. By contrast, the complexity of the WGBS libraries was lower and would have not justified a further sequencing experiment (Fig 5C). These results further suggest that WGBS is not suitable for the profiling of DNA methylation at genome-wide level in oak heartwood.

## Discussion

We tested the effect of sample preparation and tissue lysis conditions on DNA yield with the aim to maximise the amount of DNA recovered from oak sapwood and heartwood. For what concerns sample preparation, wood ground homogenisation by bead-beating in addition to burr-grinding was critical to maximise DNA yield (S1B Fig).

We showed that the lysis buffer STE 2% SDS 6% PVP and aDNA without DTT maximised DNA yield in sapwood and heartwood respectively (S1D Fig). In addition to the DNA yield, we also wanted to optimise the composition of the lysis buffer to reduce the content of PCR inhibitors in the DNA extract. We found that STE heartwood extracts do not inhibit PCR. By contrast, aDNA buffers [24] and PTB buffer [27] heartwood extracts inhibited DNA amplification (Fig 1A). Therefore, buffers composed by STE and PVP allowed to obtain DNA extracts presenting a reduced amount of PCR inhibitors and a quantity of DNA compatible with the high-throughput sequencing (S3 Table). Phenolic compounds are abundant in heartwood and reported to inhibit the amplification of DNA [2, 61]. We quantified a set of phenolic compounds and ellagitannins in inhibiting and non-inhibiting DNA extracts by UPLC QTOF-MS. Among these compounds, ellagic acid reported a 15-fold variation between inhibiting and non-inhibiting DNA extracts (Fig 1B). In addition, ellagic acid was shown to intercalate into dsDNA and thus represented a candidate for the modelling of the molecular interactions that the phenolic compounds can form with dsDNA [64].

Molecular dynamics simulations showed that ellagic acid tends to form more hydrogen bonds with the solvent in water with respect to methanol (Fig 2). In the presence of dsDNA, ellagic acid formed more interactions with dsDNA in water than in methanol, favouring minor groove locations in water and major grooves in methanol as site of approach and attachment, as shown in Fig 2. Ellagic acid also has the ability to intercalate into DNA (Fig 2), with preferential unbinding paths through the major grooves in both solvents. The reduced number of interactions observed in methanol with respect to water might explain the increased rate of PCR success observed in wood samples washed with a methanol buffer before DNA extraction observed in a previous study [14].

We tested the inhibitory effect of the subset of phenolic compounds detected by UPLC QTOF-MS using a concentration range that included or was above the values reported in the DNA extracts (Fig 1D). DNA was amplified even after the addition of ellagic acid to the reaction mix suggesting that the intercalation to dsDNA molecule, observed through molecular dynamics simulations, was not sufficient to impair PCR (Fig 1D). This assay excluded that the set of tested compounds were responsible for PCR inhibition by aDNA and PTB buffers. However, tested molecules are a small subset of the oak wood phenolic compounds so we could not exclude the idea that other compounds from the same group could inhibit PCR and that the amplification of the DNA was impaired by a combination of multiple phenolic compounds

that we detected in the heartwood DNA extracts [72]. At the same time, it cannot be excluded that other compounds, such as polysaccharides and proteins, known to inhibit PCR and likely to be present in oak heartwood, could have been responsible for the impairment of DNA amplification [61].

WGS library preparation success was dependent on the age of the sample. ∼200 years old heartwood samples (GLOR01 AD 1811–1815, 1816–1820, 1821–1830) reported a value of endogenous reads ranging from 0.13% to 0.72%. These values are similar or even higher than the ones reported for ancient oak heartwood samples considered to contain a relevant endogenous DNA content [24]. Despite that, we did not consider the amount of reads we obtained sufficient to demonstrate our ∼200 years old samples contained truly endogenous DNA. AD 1811–1830 DNA libraries were obtained from samples decontaminated by surface removal. At the same time, all the sequenced libraries prepared from samples treated with bleach, ethanol and water had a high content of endogenous DNA. These results suggest surface removal of wood samples is not sufficient to prevent contamination. Even after the optimisation of the storage conditions, sample preparation and DNA extraction, the success of WGS library preparation was limited to samples that were ∼ 100 years old, suggesting that the age of the sample is a primary limiting factor.

We then investigated the mechanism through which the age of the sample could have been linked to the success of WGS library preparation. The decay of a DNA molecule after the cell death is influenced by the environment in which the DNA is conserved and is a function of the time of conservation [69]. Decayed DNA molecules show signs of damage that can be explored through the analysis of high-throughput sequencing DNA reads. Damaged DNA fragments are expected to be short (<40bp), and have an increased cytosine deamination and depurination rate toward read ends. In our samples, cytosine deamination value did not consistently increase at read ends and reported values an order of magnitude below typical oak aDNA (Fig 4F) shown in a previous work [24]. At the same time, regions preceding DNA fragment starts were not enriched in purines excluding depurination as the cause of DNA fragmentation (Fig 4E). Despite cytosine deamination or depurination not being present at a significant rate, DNA fragments length decreased shifting from sapwood to heartwood and further reduced across heartwood samples (Fig 4A–4D). The median fragment length of the oldest sequenced sample (AD 1902–1915), 39 bp, was similar to values previously reported by ancient oak wood dated between 550 to 9,800 years [24] (Fig 4A–4D). The similarity between the fragment length of waterlogged ancient wood [24] and much younger heartwood from standing tree suggests the environment present in tree trunk might accelerates DNA degradation. We therefore concluded that DNA fragmentation, that is linked to the age of the sample, could have limited the success of WGS library preparation by forming DNA molecules too short to be compatible with the preparation of a high-throughput sequencing library. Programmed cell death and nucleic acid degradation of parenchyma cells occurring during the sapwood to heartwood transition explains the difference in DNA fragment length between these tissues [2]. Nucleases activity is unlikely to be maintained after the transition so the continued fragmentation of DNA in heartwood over time must have a different cause. We did not test if the burr-grinding and bead-beating caused the fragmentation of the DNA molecules. However, even if this effect was present, we expected the homogenisation to have contributed equally to the fragmentation of the DNA molecules in each sample. Therefore, the fragmentation of DNA we observed across the tree trunk was interpreted as a genuine, experimental artefact-free decay process occurring over time in the tree trunk.

We generated WGBS for oak sapwood and heartwood of two tree cores. The low genome coverage and depth achieved mapping the bisulfite-converted DNA reads allowed us to retrieve only a small set of informative regions that was not sufficient to characterise the DNA

methylation profile of oak heartwood. The fraction of genome covered by the bisulfite-converted DNA reads was lower than the corresponding untreated DNA reads (Fig 5A). This outcome was expected as bisulfite conversion is known to degrade DNA leading to a reduction in library complexity, as suggested by the Preseq analysis, and genome coverage (Fig 5A–5C) [56, 60, 71]. The reduction of WGBS library complexity caused by the bisulfite treatment would have impeded a significant increase in the amount of unique reads and genome coverage by the resequencing at higher depth (Fig 5C). Given the bottle-neck provided by bisulfite conversion, pooling and concentrating multiple DNA extracts would be advisable to increase the DNA fragments diversity and possibly obtain more complex libraries. The 56 informative (coverage value equal or higher than 5X) 100 bp regions that we were able to retrieve and analyse in both the considered trees correspond to a limited portion of the genome and do not allow to obtain any conclusion on the DNA methylation state of the oak sapwood or heartwood. Therefore, our data were not suitable to perform a comprehensive inference about the DNA methylation state of oak sapwood and heartwood.

## Conclusion

Overall, we reported that it is possible to extract and sequence DNA from oak heartwood samples up to ∼100 years old. The optimisation of sample preparation and DNA extraction condition allowed to obtain DNA suitable for the high-throughput sequencing. The DNA extracted from oak sapwood and heartwood was largely endogenous and allowed the profiling of the DNA molecules fragmentation and nucleotide damage over time. The fraction of the genome informative for DNA methylation analysis was limited by the low amount of DNA extracted from oak heartwood and the subsequent degradation of DNA caused by the bisulfite treatment. The complexity of the DNA reads produced by WGBS was not sufficient to characterise DNA methylation at single-base resolution. Our results suggest that the profiling of DNA methylation by WGBS in oak heartwood is likely to be limited to relatively recent samples and requires deep seqeuencing to achieve a sufficient genome coverage. Methods other than WGBS might lead to DNA libraries having higher complexity. The targeted methylome sequencing by post-bisulfite adaptor tagging (PBAT) method has been used for the profiling of DNA methylation starting from 125 pg of DNA, a quantity lower than that usually extracted from oak heartwood [73]. In PBAT, DNA is treated with bisulfite before the preparation of the library, thus preventing the breakage and loss of adapter-ligated DNA fragments. Given the reduced loss of DNA linked to the bisulfite treatment, and the low input DNA requirement, methylome sequencing by PBAT might lead to the characterisation of the oak heartwood methylation profile on a wider portion of the genome. Bisulfite-free methods for DNA methylation profiling might also be successfully applicable, given the absence of the bisuflite treatment and, as a consequence, the higher complexity of the DNA molecules that would be present in the high-throughput sequencing library [74].

## Supporting information

**S1 Fig. Preparation and extraction of DNA from oak heartwood samples.** A) Cumulative ring-with over time obtained from the dendrochronological analysis of the GLOR wood core samples collected from veteran oak trees located at the Horsepool Bottom Nature Reserve (Gloucester, England). B) DNA yield of wood ground homogenised by tissue-lyser bead-beating after (burr-grinding, bead-beating) or not homogenised (burr-grinding). Samples span the 1809-1980 time period and are obtained from the GLOR07 core. Dots corresponding to samples for which a WGS library preparation succeeded, failed or was not attempted are black, thick gray or thin gray circled respectively. C) DNA yield of wood powder washed with TNE

and methanol+CaCl2+DTT or not washed before DNA extraction [62, 63]. Samples span the 1874-1980 time period and are obtained from the GLOR07 core. Dots corresponding to samples for which a WGS library preparation succeeded, failed or was not attempted are black, thick gray or thin gray circled respectively. D) Standardised (SD) DNA yield of sapwood and heartwood extracts obtained with STE 2% SDS 6% PVP, STE 0.2% SDS 6% PVP, PTB, aDNA, aDNA without DTT buffers. E) DNA yield of GLOR07 core segments spanning the 1809-1980 time period which were air-dried and stored in silica gel or frozen after collection. Dots corresponding to samples for which a WGS library preparation succeeded, failed or was not attempted are black, thick gray or thin gray circled respectively.
(TIF)

**S1 Table. Settings of the thermocycler for the amplification of oak DNA.** Settings of the thermocycler for the PCR targeting oak sapwood and heartwood chloroplast and nuclear regions.
(XLSX)

**S2 Table. Sequences of the rbcl primers.** Sequences of the primers targeting the 203 bp region of the rbcl chloroplast gene used for the test on the amplification of sapwood and heartwood DNA.
(XLSX)

**S3 Table. Oak sapwood and heartwood WGS and WGBS sequencing metrics.** Oak sapwood and heartwood DNA reads that were generated by the sequencing of the WGS/WGBS libraries. Total reads: percentage of them that were mapped to the oak genome. Map reads: reads mapped to the oak genome. Dedup map reads: reads mapped to the oak genome and removed from PCR duplicates. Dedup nuclear map reads: reads mapped to the oak nuclear genome and removed from PCR duplicates. The percentage of reads mapped to the oak genome before the removal of PCR duplicates is not reported for the WGBS libraries as bs-seeker2, the tool used for the alignment, automatically removes the duplicated reads [54]. Seq: high-throughput sequencing library strategy.
(XLSX)

**S1 File. Experimental meta-data and measurements for 427 DNA-extracted samples.** Details on the protocol used for the preparation and the extraction of DNA from the oak wood samples; measured DNA and phenolic compound concentrations where available; outcome of the WGS library preparation where attempted.
(XLSX)

**S1 Raw images.**
(PDF)

## Acknowledgments

We thank Fabrizio Cleri (University of Lille, France) for useful discussions on DNA modelling.

## Author Contributions

**Conceptualization:** Carla Molteni, Reiner Schulz.

**Data curation:** Federico Rossi, Alessandro Crnjar, Reiner Schulz.

**Formal analysis:** Federico Rossi, Alessandro Crnjar, Reiner Schulz.

**Funding acquisition:** Carla Molteni, Reiner Schulz.

**Investigation:** Federico Rossi, Alessandro Crnjar, Federico Comitani, Rodrigo Feliciano, Leonie Jahn, George Malim, Laura Southgate, Emily Kay, Andy Moir, Reiner Schulz.

**Methodology:** Federico Rossi, Alessandro Crnjar, Federico Comitani, Rodrigo Feliciano, Leonie Jahn, George Malim, Laura Southgate, Emily Kay, Andy Moir, Logan Kistler, Ana Rodriguez Mateos, Carla Molteni, Reiner Schulz.

**Project administration:** Reiner Schulz.

**Resources:** Rebecca Oakey, Ana Rodriguez Mateos, Carla Molteni.

**Supervision:** Rebecca Oakey, Richard Buggs, Logan Kistler, Ana Rodriguez Mateos, Carla Molteni, Reiner Schulz.

**Validation:** Andy Moir, Ana Rodriguez Mateos, Carla Molteni.

**Visualization:** Federico Rossi, Alessandro Crnjar, Reiner Schulz.

**Writing – original draft:** Federico Rossi, Alessandro Crnjar, Reiner Schulz.

**Writing – review & editing:** Federico Rossi, Reiner Schulz.

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
