## [Decision Letter · Decision Letter 0]

17 Aug 2021

PONE-D-21-21916

Extraction and high-throughput sequencing of oak heartwood DNA: assessing the feasibility of genome-wide DNA methylation profiling

PLOS ONE

Dear Dr. Schulz,

Thank you for submitting your manuscript to PLOS ONE. After careful consideration, we feel that it has merit but does not fully meet PLOS ONE’s publication criteria as it currently stands. Therefore, we invite you to submit a revised version of the manuscript that addresses the points raised during the review process.

In particular, one of the two independent reviewer that has assessed you manuscript has some concerns. Please reply to their comments/concerns before resubmitting the manuscript

We look forward to receiving your revised manuscript.

Kind regards,

Emidio Albertini, Ph.D.

Academic Editor

PLOS ONE

Journal Requirements:

2. In your Methods section, please provide additional location information of the collection site, including geographic coordinates for the data set if available.

This research was funded by the King’s College London Department of Medical and Molecular Genetics, and the UK Genetics Society. This work was also supported by the King’s Together Multi and Interdisciplinary Research Scheme: Wellcome Trust Institutional Strategic Support Fund (grant reference: 204823/Z/16/Z). Sequencing was performed at the NIHR Biomedical Research Centres, London, UK. AC and CM acknowledge the UK high performance computing service ARCHER, for which access was obtained via the UKCP consortium and funded by EPSRC grant EP/P022472/1, and the UK Materials and Molecular Modelling Hub, which is partially funded by EPSRC grant EP/P020194/1 for computational resources; they also thank Fabrizio Cleri (University of Lille, France) for useful discussions on DNA modelling.”

We note that you have provided funding information within the Acknowledgements. Please note that funding information should not appear in the Acknowledgments section or other areas of your manuscript. We will only publish funding information present in the Funding Statement section of the online submission form.

“This research was funded by the King’s College London Department of Medical and Molecular Genetics, and the UK Genetics Society. This work was also supported by the King’s Together Multi and Interdisciplinary Research Scheme: Wellcome Trust Institutional Strategic Support Fund (grant reference: 204823/Z/16/Z). Sequencing was performed at the NIHR Biomedical Research Centres, London, UK. AC and CM acknowledge the UK high performance computing service ARCHER, for which access was obtained via the UKCP consortium and funded by EPSRC grant EP/P022472/1, and the UK Materials and Molecular Modelling Hub, which is partially funded by EPSRC grant EP/P020194/1 for computational resources. The funders had no role in study design, data collection and analysis, decision to publish, or preparation of the manuscript.”

7. We note that Figure 1 in your submission contain copyrighted images. All PLOS content is published under the Creative Commons Attribution License (CC BY 4.0), which means that the manuscript, images, and Supporting Information files will be freely available online, and any third party is permitted to access, download, copy, distribute, and use these materials in any way, even commercially, with proper attribution. For more information, see our copyright guidelines: http://journals.plos.org/plosone/s/licenses-and-copyright.

8. We note you have included a table to which you do not refer in the text of your manuscript. Please ensure that you refer to Table 1 and 2 in your text; if accepted, production will need this reference to link the reader to the Table.

Reviewers' comments:

Reviewer's Responses to Questions

**Comments to the Author**

1. Is the manuscript technically sound, and do the data support the conclusions?

Reviewer #1: Yes

Reviewer #2: Yes

2. Has the statistical analysis been performed appropriately and rigorously? 

Reviewer #1: I Don't Know

Reviewer #2: N/A

3. Have the authors made all data underlying the findings in their manuscript fully available?

Reviewer #1: Yes

Reviewer #2: Yes

4. Is the manuscript presented in an intelligible fashion and written in standard English?

Reviewer #1: Yes

Reviewer #2: Yes

5. Review Comments to the Author

Reviewer #1: In the study, the authors investigated protocols for the extraction of DNA, sequencing of DNA libraries and profiling of DNA methylation for oak heartwood drill cores taken from the trunks of living old trees. DNA preserved in heartwood of old trees could inform past environmental conditions and tree growth, as well as tree genetics. Although the results of the study are not surprising, the protocols should be very useful for developing technics for studying the dendrochronology and tree ancient DNA.

I have some minor questions as follows.

1. DNA fragmentation and phenolic compounds in heartwood are likely the major problems impeding the study. Is there any specific kit can remove phenolic compounds? Can larger amount of samples compensate for DNA fragmentation? Is there any pulverizer can crush more heartwood samples?

2. Since symbiotic bacteria are usually found in xylem of some living trees and wood rotten fungi are common in old tree. Is there any DNA contamination by fungi or other microorganisms in heartwood of old tree?

3. Materials are from four oak (Quercus spp.) living trees, thus I think that it is not difficult to identify the oak species name. Using “Quercus spp.” is not appropriate since Quercus are a large genus, and different Quercus species have different genome sequence.

Reviewer #2: Rossi et al. describe a procedure for DNA extraction coupled to next-generation sequencing (NGS) from heartwood and sapwood of centuries-old oaks. The authors tested three main extraction methods, identifying the best conditions. Moreover, the authors performed mass-spectrometry analysis of contaminating phenolic compounds in DNA extracts that can compromise the downstream preparation of NGS libraries. Despite being able to prepare whole-genome sequencing libraries, the limited amount and high non-oak percentage of the extracted DNA did not produced satisfying results for the assessment of DNA methylation patterns through bisulfite conversion.

Comments:

- line 82: what assay was run for quantification with the Qubit?

- line 83: it would be good to create a new table with the quantification results of the extracted DNA for each sample

- lines 170-186: please indicate what specific DNA polymerase was used for library amplification in both WGS and WGBS since it can greatly affect the yields of the library prep and the ability to amplify uracil-containing DNA. Also, please add the cat# of the reagents used.

- figures 3D, 3E, 3F: I believe that box plots/violin plots with two categories (1,0) would be more suitable for visualization.

- figures 5B (as described in the legend) missing, while 5C and D are mislabeled.

- I think that it would be more logical to present figure 2 first (e.g. MS results of the contaminants in the extracted DNA), then figure 1 (modeling of one of the contaminants..).

- I think that it would be important to know the size distribution of the extracted DNA since no fragmentation step is described in the WG(B)S library preparation protocol. It's possible (although unlikely) that HMW DNA is present in the extracted samples, but not incorporated into the final libraries because of the size. Moreover, if the DNA is extremely fragmented, the primers designed to amplify a 203-bp region of the chloroplast genome will not work efficiently or at all.

- I disagree with the WGBS library prep design used by the authors. Highly fragmented and low input samples should be processed with a post-BS (if bisulfite is chosen as conversion method) library prep strategy. This is similar to what it is usually employed for plasma cfDNA studies. Since it would be impossible, I believe, to repeat the WGBS experiments, please discuss this in the manuscript. Moreover, WGS libraries can be improved with similar single-stranded library prep methods.

6. PLOS authors have the option to publish the peer review history of their article (what does this mean?). If published, this will include your full peer review and any attached files.

Reviewer #1: No

Reviewer #2: No

---

## [Author Response · Author response to Decision Letter 0]

8 Oct 2021

Please refer to the submitted Response to Reviewers document.

---

## [Decision Letter · Decision Letter 1]

28 Oct 2021

Extraction and high-throughput sequencing of oak heartwood DNA: assessing the feasibility of genome-wide DNA methylation profiling

PONE-D-21-21916R1

Dear Dr. Schulz,

We’re pleased to inform you that your manuscript has been judged scientifically suitable for publication and will be formally accepted for publication once it meets all outstanding technical requirements.

Kind regards,

Emidio Albertini, Ph.D.

Academic Editor

PLOS ONE

Additional Editor Comments (optional):

Reviewers' comments:

Reviewer's Responses to Questions

**Comments to the Author**

1. If the authors have adequately addressed your comments raised in a previous round of review and you feel that this manuscript is now acceptable for publication, you may indicate that here to bypass the “Comments to the Author” section, enter your conflict of interest statement in the “Confidential to Editor” section, and submit your "Accept" recommendation.

Reviewer #1: (No Response)

Reviewer #2: All comments have been addressed

2. Is the manuscript technically sound, and do the data support the conclusions?

Reviewer #1: (No Response)

Reviewer #2: Yes

3. Has the statistical analysis been performed appropriately and rigorously? 

Reviewer #1: (No Response)

Reviewer #2: N/A

4. Have the authors made all data underlying the findings in their manuscript fully available?

Reviewer #1: (No Response)

Reviewer #2: Yes

5. Is the manuscript presented in an intelligible fashion and written in standard English?

Reviewer #1: (No Response)

Reviewer #2: Yes

6. Review Comments to the Author

Reviewer #1: (No Response)

Reviewer #2: The authors addressed the reviewer's comments adequately. Almost all of the requested changes have been added to the text. In addition, the authors explained why some of the reviewer's requests were not introduced.

7. PLOS authors have the option to publish the peer review history of their article (what does this mean?). If published, this will include your full peer review and any attached files.

Reviewer #1: **Yes: **Biao Jin

Reviewer #2: No

---

## [Editor Report · Acceptance letter]

9 Nov 2021

PONE-D-21-21916R1 

Extraction and high-throughput sequencing of oak heartwood DNA: assessing the feasibility of genome-wide DNA methylation profiling 

Dear Dr. Schulz:

I'm pleased to inform you that your manuscript has been deemed suitable for publication in PLOS ONE. Congratulations! Your manuscript is now with our production department. 

Kind regards, 

on behalf of

Prof. Emidio Albertini 

Academic Editor

PLOS ONE